# Radiation therapists' perceptions of thermoplastic mask use for head and neck cancer patients undergoing radiotherapy at Ocean Road Cancer Institute in Tanzania: A qualitative study

Furahini Yoram[1]*, Nazima Dharsee[1,2], Dickson Ally Mkoka[3], Khamza Maunda[1,2], Jumaa Dachi Kisukari[1,2]

1 Department of Clinical Oncology, Muhimbili University of Health and Allied Sciences, Dar es Salaam, Tanzania, 2 Academic and Research Unit, Ocean Road Cancer Institute, Dar es Salaam, Tanzania, 3 Department of Clinical Nursing, Muhimbili University of Health and Allied Sciences, Dar es Salaam, Tanzania

* furahiniyoram@gmail.com

## Abstract

### Introduction

A thermoplastic mask is the most widely used immobilization device for head and neck cancer patients undergoing radiotherapy. The radiation therapist is the staff responsible to prepare these masks and set-up the patients for treatment, a procedure that requires time, patience, and precision. An understanding of Radiation therapists' perceptions regarding thermoplastic mask use will help design interventions to address challenges encountered in its use. This study explored Radiation therapists' perceptions of thermoplastic mask use for head and neck cancer patients undergoing radiotherapy at Ocean Road Cancer Institute in Tanzania.

### Material and methods

An exploratory qualitative study design was used to explore thermoplastic mask use for head and neck cancer patients undergoing radiotherapy. Semi-structured in-depth interviews were conducted, involving fifteen Radiation therapists from Ocean Road Cancer Institute in Tanzania between March and May 2021. A thematic analysis method was used to identify themes from data scripts.

### Results

Four themes emerged that reflected radiation therapists' perceptions of thermoplastic mask use for head and neck cancer immobilization among patients undergoing radiotherapy. Emerged themes were (1) Perceived benefits and limitations of thermoplastic mask use, (2) Refresher training and supervision requirements for effective use, (3) Proper storage for quality maintenance, and (4) Increased financial support and proper budgeting.

**Data Availability Statement:** All relevant data are within the paper and its Supporting information files.

**Funding:** The author(s) received no specific funding for this work.

**Competing interests:** The authors have declared that no competing interests exist.

## Conclusion

Participants perceived better patient immobilization with a thermoplastic mask use. However, too often recycling of thermoplastic masks and the long waiting time between thermoplastic mask preparation and treatment delivery limits their effective use. For efficient use of thermoplastic masks, there is a need for Radiation therapists' refresher training and proper supervision, improving the storage system and increasing financial support for procuring new thermoplastic masks.

## Introduction

Head and neck cancers (HNC) are usually managed by multimodality approaches such as surgery, chemotherapy, and radiotherapy (RT). Nearly 75% of HNC require RT as a mode of treatment for curative or palliative intent [1]. The efficiency of RT for HNC depends largely on proper patient immobilization. This is because anatomically the head and neck contain important critical structures (Organs at risk) in proximity to each other [2]. The major goals of the immobilization system are to restrict patient motion during treatment and to minimize positioning errors by ensuring daily reproducibility of the patient treatment position throughout the RT course [3–5].

A thermoplastic mask is the most widely used immobilization device for HNC patients undergoing RT. This is a perforated plastic sheet that becomes soft when placed in warm water and it can be moulded to conform to the patient's anatomy [6, 7]. Different thermoplastic mask types are used depending on the site of the disease. S-type thermoplastic masks cover the head, neck, and shoulders while U-type thermoplastic masks cover the head only (Fig 1).

A radiation therapist (RTT) is responsible for RT treatment preparation including the construction of a thermoplastic mask in the mould room, positioning the patient during computed tomography (CT) simulation, treatment verification, and delivering radiation dose to the patient during treatment [8]. During thermoplastic mask preparation, RTTs are supposed to work as a team and maintain proper documentation of the treatment parameters [9].

Thermoplastic masks require sufficient space in the RT department for safe storage during the entire treatment. Furthermore, RTTs who prepare these masks and set up the patients for treatment need to be adequately skilled and experienced in the procedure which can be challenging and time-consuming. A positive attitude facilitates the efficient use of the thermoplastic

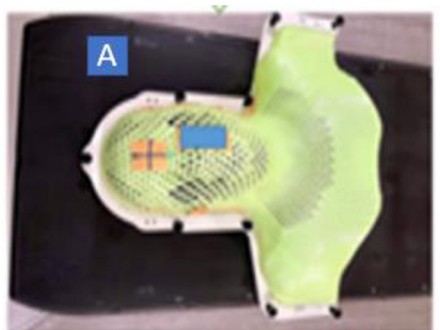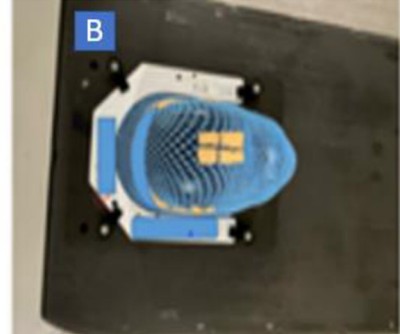

**Fig 1. Different thermoplastic mask types (A) S-type (B) U-type.**

mask in the RT department. An understanding of RTTs' perceptions regarding thermoplastic mask use will help design interventions to address challenges encountered in its use, especially in resource-limited settings. This study aimed at exploring RTTs' perceptions on the use of the thermoplastic mask for HNC patients undergoing RT at Ocean Road Cancer Institute in Tanzania. This will help to improve RT practice regarding thermoplastic mask use.

## Material and methods

### Study design

An exploratory qualitative study was conducted using in-depth interviews to explore RTTs' perceptions of the use of thermoplastic masks as an immobilization device for HNC patients undergoing RT at Ocean Road Cancer Institute.

### Study setting

The study took place at Ocean Road Cancer Institute in Dar es Salaam, Tanzania. It is a national referral cancer center from Tanzania receiving cancer patients from all over the country and abroad who are histologically confirmed with cancer. It offers radiotherapy (RT), chemotherapy, brachytherapy, Nuclear Medicine, and palliative care services. Normally the RT for HNC patients takes up to 33 days (7 weeks) of the treatments on the 5 consecutive days of a week for curative cases.

Management of HNC patients involves a multidisciplinary team including Clinical/Radiation Oncologists, Radiation therapists (RTTs), Medical Physicists, Nurses, and other supporting staff. Ocean Road Cancer Institute has thirty (30) employed RTTs who each received training on administering RT to all cancer sites including HNC. Their main responsibilities are treatment preparation, radiotherapy planning, and delivery in the Institute. During treatment preparation, they prepare different immobilization devices including thermoplastic masks for HNC patients. Due to a large number of patients and limited resources, sometimes thermoplastic masks were recycled to immobilize HNC patients.

### Participants

This study involved fifteen (15) experienced RTTs working in the RT department at Ocean Road Cancer Institute. A purposive sampling strategy was used to recruit RTTs with various working experiences in the RT department. RTTs who were selected were those responsible for preparing HNC patients' treatment and delivering radiation doses during the entire course of RT.

### Data collection methods and tools

Data collection was done using a semi-structured interview guide developed based on a literature review as well as the clinical experiences of researchers working in a department with a high workload of patients and limited resources. The guide was pre-tested and after the final revision, the guide included questions that explored benefits and limitations, challenges, and possible solutions to overcome the stated challenges associated with thermoplastic mask use for HNC patients. These questions were followed with probe questions based on participants' responses.

Fifteen (15) In-depth interviews were conducted by the researchers between March and May 2021. Interviews took time between 20 and 30 minutes to complete. Interviews were conducted in Swahili which is a national language and is usually used for communication in the Institute.

Principle of data saturation guided to reach a sample size of 15 participants. This means, the sample size was not predetermined, instead, the interviews stopped when the researchers noticed the repetition of the earlier gained information [10]. All the interviews were conducted in a special room prepared for the interviews to maintain participants' privacy. Confidentiality was guaranteed by referring participants using numbers not names during interviews. Before the interviews and audio recording, participants were requested for written informed consent. Interviews were audio-recorded to gather all information provided by participants and notes were taken using notebooks to get some non-verbal data that were missed from the recorder.

## Data analysis

The thematic analysis approach was used as described by Braun and Clark [11]. The analysis followed six steps of thematic analysis [12, 13]. First, Audio-recorded interviews in Swahili language were transcribed verbatim immediately after the completion of the interviews by the authors. The Swahili transcripts were then translated into the English language by the first author and a quality check was done to confirm the accuracy of the transcription and translations by other authors. Reading and re-reading of transcribed interviews were done by all authors to become familiar with the data. Second, lists of initial ideas were generated by two authors from transcripts and cross-checked by other authors. Third, initial codes were developed and organized into themes. Fourth, themes were reviewed by all authors. Fifth, each theme was refined by all authors. Finally, the report was produced with codes and themes that were presented with the support of succinct quotes.

## Ethics approval and consent to participate

Ethical approval to conduct this study was obtained from the Institutional Review Board of the Muhimbili University of Health and Allied Sciences and permission to conduct this study was obtained from the Institute Academics, Research, Publications, and Ethical Committee of Ocean Road Cancer Institute. Participants provided written informed consent for interviews and recording.

## Results

### Participant characteristics

The 15 participants were aged between 27 and 49 years. The majority (80%) of the participants were male. All the participants were holders of bachelor's degrees in Radiation therapy. Nine (9) participants had professional experience of less than 5 years of practice in the RT department.

### Themes

Four themes emerged that describe participants' perceptions of thermoplastic mask use for the treatment of HNC. Emerged themes were Perceived benefits and limitations; refresher training and supervision requirements for effective use; proper storage for quality maintenance and increased financial support and proper budgeting. Table 1 indicates emerging themes with corresponding codes.

### Perceived benefits and limitations

Participants in this study highlighted some benefits of using thermoplastic masks during the treatment of HNC patients. Participants acknowledged that the use of a thermoplastic mask as an immobilization device maintains treatment position throughout the RT course and also it

**Table 1. Themes and corresponding codes for RTTs' perceptions of thermoplastic mask use.**

| Corresponding codes | Themes |
|---|---|
| Ensures easy reproducibility | Perceived benefits and limitations |
| It is user friendly | |
| Quality decreases as you recycle | |
| Long waiting time affects its use | |
| Differences in mask preparation | Refresher training and supervision requirements for effective use |
| Refresher training is helpful | |
| Following guidelines for effective use | |
| Frequent supervision from an expert is required | |
| The storage place is supposed to be big | Proper storage for quality maintenance |
| Use of logbook for easy identification | |
| Having a sufficient shelving | |
| Mask requires safe storage | |
| Increasing budget for thermoplastic masks | Increased financial support and proper budgeting |
| Using a new thermoplastic mask is best | |
| Limited new thermoplastic mask availability | |
| Minimizing thermoplastic mask recycling | |

ensures easy reproducibility. This was recognized as contributing to effective treatment which minimizes the radiation dose to the adjacent structures since the head and neck have many structures close to each other. A thermoplastic mask is also identified as being a user friendly because it is easy to construct and use to the extent that it can be prepared and used within a day as stated by one respondent:

> "*That device is used to maintain a patient in a fixed position to ensure the correct delivery of radiation dose to the intended area and it is easy to construct. In our department, the patient can come today, you construct a thermoplastic mask today and start using it today during treatment*".

> (*RTT 6*)

Despite the benefits of the thermoplastic mask, some limitations have been explained by the participants. They perceived that the quality of the thermoplastic mask decreases as you recycle (remould) compared to the new one due to the loss of its elasticity property that results from repeated use.

> "*. . .the more you recycle* (*remould*) *thermoplastic mask, the more it loses its quality and increases the possibility of geometrical misses and become unfit to the patient*".

> (*RTT 3*)

Participants reported that a delay in the start of RT after CT simulation would affect the immobilization status of the mask making it unfit during treatment. This could decrease treatment efficiency and also make the mask uncomfortable for the patient as narrated by one participant.

> "*We have been getting complaints from the patients that the thermoplastic mask compresses or is unfit than it's supposed to be*". *This occurs when there is a long waiting time between the*

*thermoplastic mask construction and the treatment delivery.* "*Usually more than two weeks is enough to lose its immobilization status and it starts compressing the patient or becoming unfit*".

(*RTT 2*)

## Refresher training and supervision requirements for effective use

Participants also suggested the need for refresher training and supervision among RTTs as some of the thermoplastic masks were seen with improper immobilization during treatment due to the differences in mask preparation among RTTs. Since they have basic training, doing refresher training will help them to improve their competency and skills in the effective use of a thermoplastic mask.

"*I think what is best first is to have continuing medical education for staff and refresher training. So if there is frequent refresher training on properly using a thermoplastic mask, it will help remind the staff of its proper use*".

(*RTT 1*)

Participants also pointed out the need for proper supervision to reinforce their competency during thermoplastic mask preparation. This can be achieved by having a protocol or standard operating procedures that every RTT has to follow and there should be a supervisor to make follow up on its effective implementation in the department.

"*……..the big issue is lack of supervision because if it's basic training every staff has got that. What is important is proper supervision in implementing protocol or standard operating procedures*".

(*RTT 12*)

## Proper storage for quality maintenance

Participants expressed their concern regarding the size of the storage room for keeping thermoplastic masks. Concerns included a lack of enough space to store thermoplastic masks in the department. The absence of sufficient storage shelving in a room resulted in thermoplastic masks being put on top of each other. Lack of these may transmit infections if there is no proper hygiene and will even distort their shapes.

"*To my understanding, it is required to have sufficient storage shelving which helps to properly keep thermoplastic masks, but in our setting, we don't have enough storage shelving. What we do is put the thermoplastic mask on top of one another. There is a possibility that the one which is beneath others gets deformed. If this happens, it becomes unfit for the planned patient*".

(*RTT 9*)

Participants also described that the use of a logbook could help in easy identification of a thermoplastic mask. Additionally, the absence of a logbook is considered to be the reason for the delay of treatment in using much time searching for thermoplastic masks as stated by one participant.

"*Searching for the thermoplastic mask is the main challenge, there should be a proper arrangement by having a logbook for easy identification*".

(*RTT 5*)

## Increased financial support and proper budgeting

Several participants discussed the issue of thermoplastic mask availability in the department as a challenge due to limited funds for procuring a new thermoplastic mask that resulted in remoulding previously used thermoplastic masks. Participants voiced for each patient to use a new mask, and this will require adequate planning and budgeting by the radiotherapy department.

"*Increase a budget that will enable us all the time to have new thermoplastic masks instead of re-moulding used thermoplastic masks which are inefficient*".

(*RTT 7*)

## Discussion

The objective of this study was to explore RTTs' perceptions of thermoplastic mask use for HNC patients' immobilization during RT. In this study, participants reported that a thermoplastic mask ensures easy reproducibility of the treatment setups. Studies have reported that the use of a thermoplastic mask ensures the accurate delivery of the treatment to the intended area by reducing daily setup variations [14, 15].

Thermoplastic mask use is limited to several factors as reported in this study. Re-moulding or recycling a thermoplastic mask decreases its quality compared to the new mask. This is because it forms folds that maximize the dose to patients' skin especially when not well moulded in the first use. This affects its immobilization property during treatment. It has been reported that recycling a thermoplastic mask too often results in poor immobilization due to loss of rigidity [16]. Also, the long waiting time between thermoplastic mask construction and treatment delivery limits the effective use of thermoplastic masks. Changes in the size of externally growing tumours during this period make the mask unfit for the patient which sometimes necessitates repeating the whole process of mould room and treatment planning [17].

This study revealed differences in RTTs' understanding of the thermoplastic mask preparation process. Participants reported the need for refresher training and proper supervision of RTTs working in the RT department. All the RTTs got good training during their studies, however, due to the advancement of RT techniques and procedures, there might be some deficiencies in their competency. Competency is needed and this can be improved through regular training. Likewise, supervision is required in the RT department that will help RTTs adhere to departmental protocol.

There is a need for proper storage of thermoplastic masks for quality maintenance. This was reported as one of the challenges in the study setting. Due to the limited storage system, sometimes there was a misplacement of thermoplastic masks. It has been recommended that treatment and simulator rooms are supposed to be big enough with sufficient storage shelving so as to store different immobilization devices [18]. Furthermore putting thermoplastic masks on top of each other might distort their shapes and even transmit infection if there is no proper hygiene. Therefore, proper storage is required to prevent thermoplastic mask distortion [16].

Our findings also revealed that there is a shortage of thermoplastic mask availability. This could be associated with a large number of patients who require thermoplastic masks during RT. The study further found limited funding that affected the availability of thermoplastic masks in the RT department. This led to the remoulding of the used thermoplastic masks as reported in this study. This is unavoidable in limited-resource settings due to financial constraints as compared to the developed countries which usually discard them after patients complete their treatment [19]. However, increasing the budget to ensure quality services to the patients is a proper way [20].

## Trustworthiness of the study

The concept of trustworthiness is assessed by using four criteria; credibility, transferability, confirmability and dependability [13]. In this study, credibility was ensured through the inclusion of participants with various experiences in managing HNC patients. Also, data collection was conducted by authors who have vast experience in managing patients undergoing radiotherapy. To enhance transferability, a thick description of the study context was provided so that findings could be transferred to a site with a similar context. To enhance confirmability, authors with different educational and experience backgrounds participated in the whole process of the study. They derived and interpreted the findings from the data collected from the study participants and not from their ideas. The findings were supported by codes and quotes. Dependability was enhanced through good documentation of the research process that can be traced.

## Study limitation

This study was done in one cancer center in Tanzania, hence these findings might have limited generalizability. However, many cancer centers in developing countries share similar challenges. Therefore, the findings from this study are transferable to other similar settings.

## Conclusion

Participants perceived better patient immobilization with a thermoplastic mask use. However, too often recycling of thermoplastic masks and the long waiting time between thermoplastic mask preparation and treatment delivery limits their effective use. There is also a need for RTTs' refresher training and proper supervision, improving the storage system and increasing financial support for procuring new thermoplastic masks.

To facilitate training and proper supervision, the RT department should have at least a monthly meeting to discuss challenges arising in the department and have proper supervision of departmental protocol so as to improve the proper use of masks. Also, there should be the involvement of RTTs during the design of simulation and treatment rooms so as to ensure proper storage systems for thermoplastic masks. The findings also suggest the use of new thermoplastic masks as well as minimizing waiting time as the best way to properly immobilize HNC patients.

## Supporting information

**S1 File.**
(ZIP)

## Acknowledgments

The authors wish to acknowledge all participants for their valuable time. We also wish to thank the Muhimbili University of Health and Allied Sciences for supporting this study and the Ocean Road Cancer Institute for providing permission to conduct this study.

## Author Contributions

**Conceptualization:** Furahini Yoram, Nazima Dharsee, Dickson Ally Mkoka, Khamza Maunda, Jumaa Dachi Kisukari.

**Formal analysis:** Furahini Yoram, Nazima Dharsee, Dickson Ally Mkoka.

**Methodology:** Furahini Yoram, Nazima Dharsee, Dickson Ally Mkoka, Khamza Maunda, Jumaa Dachi Kisukari.

**Supervision:** Furahini Yoram, Nazima Dharsee, Dickson Ally Mkoka, Khamza Maunda, Jumaa Dachi Kisukari.

**Writing – original draft:** Furahini Yoram.

**Writing – review & editing:** Furahini Yoram, Nazima Dharsee, Dickson Ally Mkoka, Khamza Maunda, Jumaa Dachi Kisukari.

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
