## [Decision Letter · Decision Letter 0]

31 Oct 2022

PONE-D-22-23013Radiation therapists’ perceptions of thermoplastic mask use for head and neck cancer patients undergoing radiotherapy at Ocean Road Cancer Institute in Tanzania: A qualitative studyPLOS ONE

Dear Dr. Yoram,

Thank you for submitting your manuscript to PLOS ONE. After careful consideration, we feel that it has merit but does not fully meet PLOS ONE’s publication criteria as it currently stands. Therefore, we invite you to submit a revised version of the manuscript that addresses the points raised during the review process.

The aim of the manuscript and its usefulness should be clearly defined. Additionally, the manuscript presents several methodological problems, pointed out by the reviewers, that must be carefully resolved before it can be considered for publication.

We look forward to receiving your revised manuscript.

Kind regards,

Paula Boaventura, PhD

Academic Editor

PLOS ONE

Journal Requirements:

Reviewers' comments:

Reviewer's Responses to Questions

**Comments to the Author**

1. Is the manuscript technically sound, and do the data support the conclusions?

Reviewer #1: Yes

Reviewer #2: Partly

Reviewer #3: No

Reviewer #4: Yes

2. Has the statistical analysis been performed appropriately and rigorously? 

Reviewer #1: N/A

Reviewer #2: N/A

Reviewer #3: No

Reviewer #4: I Don't Know

3. Have the authors made all data underlying the findings in their manuscript fully available?

Reviewer #1: Yes

Reviewer #2: No

Reviewer #3: Yes

Reviewer #4: Yes

4. Is the manuscript presented in an intelligible fashion and written in standard English?

Reviewer #1: Yes

Reviewer #2: Yes

Reviewer #3: Yes

Reviewer #4: Yes

5. Review Comments to the Author

Reviewer #1: Dear Authors,

Congratulations with submission of the article Radiation therapists’ perceptions of thermoplastic mask use for HNC - A

qualitative study. The article describes an important issue regarding the perception of radiation therapists and the use of thermoplastic masks. Some minor changes:

1. line 104: "patients" should be replaced by "RTT" or " participants"

2. Could you suggest more practical things that could be improved e.g. giving education, 5 times a year? creating protocols? What would be the next step?

3. For future: It would be interesting to see a follow-up of the situations, have things improved? what interventions work and what do not?

Overall, an important and interesting topic.

Reviewer #2: The objective should have been better defined.

Academic training and the number of years of professional experience should be included.

It does not mention who carried out the interviews and content analysis.

The conclusions are in part aligned with the objective.

The study is limited to a single location, which makes this a case study, with all the limitations that this type of study entails.

Reviewer #3: The article „Radiation therapists’ perceptions of thermoplastic mask use for head and neck cancer patients undergoing radiotherapy at Ocean Road Cancer Institute in Tanzania: A qualitative study” is interesting especially due to its setting in a low-resource setting. However, I think the novelty of the study should be emphasized, especially since the use of thermoplastic masks in ENT cancers has become standard. Additionally, I have several questions/comments, listed below.

1. Who designed the questionnaire? How were the questions chosen?

2. The authors refer to RTTs as radiation specilists. Does this mean they are all physicians? Or were there also nurses and physicists involved in the study?

3. Also in the Material and Method section, I think more details should be offered. How were the data analyzed? What are the six steps used for analysis? Because to me it seems that only a summary of the interviews is provided

4. The authors refer to the concept of recycling the thermoplastic masks. I think more details should be offered on the subject, especially since this is not standard

5. In the Discussions section, the authors state that „This study revealed significant differences in participants’ understanding of the thermoplastic mask preparation process.” Where was this detailed in the Results section?

Minor comments

Please revise: „This is a perforated plastic sheet that is soaked in warm water which then becomes soft and flexible, moulded on the patient’s face, cools after a certain time and becomes hard and finally keeps a final shape of the patient’s anatomy (6,7)”, „During thermoplastic mask preparation, RTT explains in detail the procedure to the patient, position the patient, and perform quick and accurate construction of thermoplastic mask by adhering to the manufacturer’s guidelines while working as a team and maintaining proper documentation (9).”, „This is because of avoiding its loss, easy identification during treatment, shape maintenance and prevention of infection transmission.”

Reviewer #4: Study seems to address the key routine issues and conducted well. However,I have few queries as below

* Experience and Knowledge level of the RTTs involved( Aged 27-49 years) is missing

* internal consistency reliability test and coherence of the Thematic analysis is missing

Kindly address these queries

6. PLOS authors have the option to publish the peer review history of their article (what does this mean?). If published, this will include your full peer review and any attached files.

Reviewer #1: No

Reviewer #2: No

Reviewer #3: No

Reviewer #4: No

---

## [Author Response · Author response to Decision Letter 0]

5 Nov 2022

Academic editor comments

1. Please ensure that your manuscript meets PLOS ONE’s style requirements, including those for file naming. The PLOS ONE style templates

Authors response - Thank you. This has been corrected in a revised manuscript

2. Please provide additional details regarding participant consent.

Authors response - Thank you. Additional details have been added in a revised manuscript

3. In your Data Availability statement, you have not specified where the minimal data set underlying the results described in your manuscript can be found.

Authors response - Thank you. This has been specified

4. Your ethics statement should only appear in the Methods section of your manuscript.

Authors response - Thank you. An ethical statement has been moved to the methods section

Reviewer 1 comments

1. line 104: "patients" should be replaced by "RTT" or " participants"

Authors response - Thank you for this observation

2. Could you suggest more practical things that could be improved e.g. giving education, 5 times a year? creating protocols? What would be the next step?

Authors response - Thank you. This has been added in the conclusion section 

3. For future: It would be interesting to see a follow-up of the situations, have things improved? what interventions work and what do not?

Authors response - Thank you. We will make a follow-up.

Reviewer 2 comments

1. The objective should have been better defined.

Authors response - Thank you. This has been revised

2. Academic training and the number of years of professional experience should be included.

Authors response - Thank you. This has been included

3. It does not mention who carried out the interviews and content analysis.

Authors response - Thank you. This has been added to the data collection and analysis section

4. The conclusions are in part aligned with the objective

Authors response - Thank you. This has been revised.

5. The study is limited to a single location, which makes this a case study, with all the limitations that this type of study entails.

Authors response - Thank you for your comment. It is true this is a major limitation for the qualitative studies whose results were not meant for generalizability but rather to explore in-depth views of the study participants about the thermoplastic mask use in the study setting. However, the findings could be transferred to settings with a similar context.

Reviewer 3 comments

1. The article „Radiation therapists’ perceptions of thermoplastic mask use for head and neck cancer patients undergoing radiotherapy at Ocean Road Cancer Institute in Tanzania: A qualitative study” is interesting especially due to its setting in a low-resource setting. However, I think the novelty of the study should be emphasized, especially since the use of thermoplastic masks in ENT cancers has become standard. Additionally, I have several questions/comments, listed below.

Authors response - Thank you for your comment. We have tried to work on it

2. Who designed the questionnaire? How were the questions chosen?

Authors response - The questionnaire was designed by the authors based on clinical experience with a high workload setting and limited resources. The statement has been added to the revised manuscript

3. The authors refer to RTTs as radiation specilists. Does this mean they are all physicians? Or were there also nurses and physicists involved in the study?

Authors response - Thank you for this good question. RTTs are not physicians, they are technologists working in a radiotherapy department. They are neither nurses nor physicists

4. Also in the Material and Method section, I think more details should be offered. How were the data analyzed? What are the six steps used for analysis? Because to me it seems that only a summary of the interviews is provided

Authors response - Thank you for this comment. This has been well explained in a revised manuscript

5. The authors refer to the concept of recycling the thermoplastic masks. I think more details should be offered on the subject, especially since this is not standard

Authors response - Thank you. More details have been offered in a study setting section

6. In the Discussions section, the authors state that „This study revealed significant differences in participants’ understanding of the thermoplastic mask preparation process.” Where was this detailed in the Results section?

Authors response - Thank you for pointing out this error. This sentence has been corrected.

7. Minor comments

Please revise: „This is a perforated plastic sheet that is soaked in warm water which then becomes soft and flexible, moulded on the patient’s face, cools after a certain time and becomes hard and finally keeps a final shape of the patient’s anatomy (6,7)”, „During thermoplastic mask preparation, RTT explains in detail the procedure to the patient, position the patient, and perform quick and accurate construction of thermoplastic mask by adhering to the manufacturer’s guidelines while working as a team and maintaining proper documentation (9).”, „This is because of avoiding its loss, easy identification during treatment, shape maintenance and prevention of infection transmission.”

Authors response - Thank you so much. We agree that these sentences are not clear. These sentences have been revised

Reviewer 4 comments

1. Experience and Knowledge level of the RTTs involved (Aged 27-49 years) is missing

Authors response - Thank you. This has been added

2. internal consistency reliability test and coherence of the Thematic analysis is missing

Authors response - Thank you. This has been added

---

## [Decision Letter · Decision Letter 1]

28 Nov 2022

PONE-D-22-23013R1

Radiation therapists’ perceptions of thermoplastic mask use for head and neck cancer patients undergoing radiotherapy at Ocean Road Cancer Institute in Tanzania: A qualitative study

PLOS ONE

Dear Dr. Yoram,

Thank you for submitting your manuscript to PLOS ONE. After careful consideration, we feel that it has merit but does not fully meet PLOS ONE’s publication criteria as it currently stands. Therefore, we invite you to submit a revised version of the manuscript that addresses the points raised during the review process.

We look forward to receiving your revised manuscript.

Kind regards,

Paula Boaventura, PhD

Academic Editor

PLOS ONE

Journal Requirements:

Additional Editor Comments:

Please, carefully address the earlier queries of Reviewer 4, which were not addressed in the revised version of the manuscript (although this was stated in the authors reviewers reply). The discussion needs to be revised according to the Reviewer 2 comments.

Reviewers' comments:

Reviewer's Responses to Questions

**Comments to the Author**

1. If the authors have adequately addressed your comments raised in a previous round of review and you feel that this manuscript is now acceptable for publication, you may indicate that here to bypass the “Comments to the Author” section, enter your conflict of interest statement in the “Confidential to Editor” section, and submit your "Accept" recommendation.

Reviewer #1: All comments have been addressed

Reviewer #2: All comments have been addressed

Reviewer #4: (No Response)

2. Is the manuscript technically sound, and do the data support the conclusions?

Reviewer #1: Yes

Reviewer #2: Partly

Reviewer #4: Yes

3. Has the statistical analysis been performed appropriately and rigorously? 

Reviewer #1: N/A

Reviewer #2: N/A

Reviewer #4: Yes

4. Have the authors made all data underlying the findings in their manuscript fully available?

Reviewer #1: Yes

Reviewer #2: Yes

Reviewer #4: Yes

5. Is the manuscript presented in an intelligible fashion and written in standard English?

Reviewer #1: Yes

Reviewer #2: Yes

Reviewer #4: Yes

6. Review Comments to the Author

Reviewer #1: Dear authors,

I do not have any further comments to the article.

Kind regards

Reviewer #2: Most of the reviewers' recommendations were carried out.

Standardize the type of parentheses in the Bibliographic References, straight or curved (according to the rules of the publication).

The Discussion could be more accomplished, expression as "This study revealed significant differences in participants’ understanding of the thermoplastic mask preparation process" (l248), are "too strong".

There is a need for these studies in their conclusions to contain implications and to contribute to the improvement of practices.

Reviewer #4: (No Response)

7. PLOS authors have the option to publish the peer review history of their article (what does this mean?). If published, this will include your full peer review and any attached files.

Reviewer #1: No

Reviewer #2: No

Reviewer #4: **Yes: **SRINIDHI G CHANDRAGUTHI

---

## [Author Response · Author response to Decision Letter 1]

6 Dec 2022

1. Journal requirements

Response: Thank you. All the references have been reviewed 

2. Additional editor comments

Please, carefully address the earlier queries of Reviewer 4, which were not addressed in the revised version of the manuscript (although this was stated in the authors reviewers reply). 

i. Experience and Knowledge level of the RTTs involved (Aged 27-49 years) is missing

Response: Thank you for this comment. This has been revised in the result section of the revised manuscript line 133-135

ii. Internal consistency reliability test and coherence of the Thematic analysis is missing

Response: Thank you for this comment.

I understand that reliability test and coherence is very important. However, in qualitative studies, this is described as how the trustworthiness of the study will be realized. There are four criteria to consider; credibility, transferability, confirmability and dependability. This was not included in the earlier version of the manuscript, but due to its importance, it has been included in the revised manuscript. Line 245-255

3. Reviewer 2

i. The Discussion could be more accomplished, expression as "This study revealed significant differences in participants’ understanding of the thermoplastic mask preparation process" (l248), are "too strong".

Response: Thank you so much for this observation. This has been revised in a revised manuscript line 225

ii. There is a need for these studies in their conclusions to contain implications and to contribute to the improvement of practices.

Response: Thank you for this observation. We have revised the conclusion section to include the implications of the study and how the findings will improve the practice.

Line 260-271

4. Addition to the manuscript

Study limitation section

Response: We have added this section in our revised manuscript to clear any doubts from our readers as also asked by reviewer 2. Line 256-259

---

## [Decision Letter · Decision Letter 2]

3 Jan 2023

PONE-D-22-23013R2Radiation therapists’ perceptions of thermoplastic mask use for head and neck cancer patients undergoing radiotherapy at Ocean Road Cancer Institute in Tanzania: A qualitative studyPLOS ONE

Dear Dr. Yoram,

Thank you for submitting your manuscript to PLOS ONE. After careful consideration, we feel that it has merit but does not fully meet PLOS ONE’s publication criteria as it currently stands. Therefore, we invite you to submit a revised version of the manuscript that addresses the points raised during the review process.

The manuscript has greatly improved and is almost ready for publication. Please make the last minor changes proposed by the reviewer.

We look forward to receiving your revised manuscript.

Kind regards,

Paula Boaventura, PhD

Academic Editor

PLOS ONE

Journal Requirements:

Reviewers' comments:

Reviewer's Responses to Questions

**Comments to the Author**

1. If the authors have adequately addressed your comments raised in a previous round of review and you feel that this manuscript is now acceptable for publication, you may indicate that here to bypass the “Comments to the Author” section, enter your conflict of interest statement in the “Confidential to Editor” section, and submit your "Accept" recommendation.

Reviewer #2: All comments have been addressed

2. Is the manuscript technically sound, and do the data support the conclusions?

Reviewer #2: Yes

3. Has the statistical analysis been performed appropriately and rigorously? 

Reviewer #2: Yes

4. Have the authors made all data underlying the findings in their manuscript fully available?

Reviewer #2: Yes

5. Is the manuscript presented in an intelligible fashion and written in standard English?

Reviewer #2: Yes

6. Review Comments to the Author

Reviewer #2: Minor changes:

(Material and Methods, pg5, second paragraph) rephrase "...so as to immobilize HNC patients."

(discussion, pg12, third paragraph) remove "significant"

(References) reformulate 2.

7. PLOS authors have the option to publish the peer review history of their article (what does this mean?). If published, this will include your full peer review and any attached files.

Reviewer #2: No

---

## [Author Response · Author response to Decision Letter 2]

4 Jan 2023

Reviewer comment 1: (Material and Methods, pg5, second paragraph) rephrase "...so as to immobilize HNC patients."

Response: Thank you for this comment. This has been rephrased in the material and methods section of the revised manuscript line 89

Comment 2: (Discussion, pg12, third paragraph) remove "significant"

Response: Thank you for this comment. This has been removed in the revised manuscript line 225

Comment 3: (References) reformulate 2.

Response: Thank you for this comment. Reference 2 has been reformulated in the revised manuscript. It has been revised to adhere chapter referencing style. Line 282-284

---

## [Decision Letter · Decision Letter 3]

9 Feb 2023

Radiation therapists’ perceptions of thermoplastic mask use for head and neck cancer patients undergoing radiotherapy at Ocean Road Cancer Institute in Tanzania: A qualitative study

PONE-D-22-23013R3

Dear Dr. Yoram,

We’re pleased to inform you that your manuscript has been judged scientifically suitable for publication and will be formally accepted for publication once it meets all outstanding technical requirements.

Kind regards,

Hussein ALMasri

Academic Editor

PLOS ONE

Additional Editor Comments (optional):

Dear Dr. Furahini Yoram,

It is a pleasure to accept your manuscript entitled "Radiation therapists’ perceptions of thermoplastic mask use for head and neck cancer patients undergoing radiotherapy at Ocean Road Cancer Institute in Tanzania: A qualitative study

" in its current form for publication in PLOS ONE. The comments of the referees who reviewed your manuscript are included at the bottom of this letter.

Reviewers' comments:

Reviewer's Responses to Questions

**Comments to the Author**

1. If the authors have adequately addressed your comments raised in a previous round of review and you feel that this manuscript is now acceptable for publication, you may indicate that here to bypass the “Comments to the Author” section, enter your conflict of interest statement in the “Confidential to Editor” section, and submit your "Accept" recommendation.

Reviewer #2: All comments have been addressed

2. Is the manuscript technically sound, and do the data support the conclusions?

Reviewer #2: Yes

3. Has the statistical analysis been performed appropriately and rigorously? 

Reviewer #2: N/A

4. Have the authors made all data underlying the findings in their manuscript fully available?

Reviewer #2: Yes

5. Is the manuscript presented in an intelligible fashion and written in standard English?

Reviewer #2: Yes

6. Review Comments to the Author

Reviewer #2: (No Response)

7. PLOS authors have the option to publish the peer review history of their article (what does this mean?). If published, this will include your full peer review and any attached files.

Reviewer #2: No

---

## [Editor Report · Acceptance letter]

14 Feb 2023

PONE-D-22-23013R3 

Radiation Therapists’ perceptions of thermoplastic mask use for head and neck cancer patients undergoing radiotherapy at Ocean Road Cancer Institute in Tanzania: A qualitative study 

Dear Dr. Yoram:

I'm pleased to inform you that your manuscript has been deemed suitable for publication in PLOS ONE. Congratulations! Your manuscript is now with our production department. 

Kind regards, 

on behalf of

Dr. Hussein ALMasri 

Academic Editor

PLOS ONE